# Language experience shapes predictive coding of rhythmic sound sequences

Piermatteo Morucci[1,2]*, Sanjeev Nara[2,3], Mikel Lizarazu[2], Clara Martin[2,4], Nicola Molinaro[2,4]

[1]Department of Fundamental Neurosciences, University of Geneva, Geneva, Switzerland; [2]Basque Center on Cognition, Brain and Language, Donostia-San Sebastian, Spain; [3]Mathematical Institute, Department of Mathematics and Computer Science, Physics, Geography, Liebig-Universität Gießen, Gießen, Germany; [4]Ikerbasque, Basque Foundation for Science, Bilbao, Spain

**Abstract** Perceptual systems heavily rely on prior knowledge and predictions to make sense of the environment. Predictions can originate from multiple sources of information, including contextual short-term priors, based on isolated temporal situations, and context-independent long-term priors, arising from extended exposure to statistical regularities. While the effects of short-term predictions on auditory perception have been well-documented, how long-term predictions shape early auditory processing is poorly understood. To address this, we recorded magnetoencephalography data from native speakers of two languages with different word orders (Spanish: functor-initial vs Basque: functor-final) listening to simple sequences of binary sounds alternating in duration with occasional omissions. We hypothesized that, together with contextual transition probabilities, the auditory system uses the characteristic prosodic cues (duration) associated with the native language's word order as an internal model to generate long-term predictions about incoming non-linguistic sounds. Consistent with our hypothesis, we found that the amplitude of the mismatch negativity elicited by sound omissions varied orthogonally depending on the speaker's linguistic background and was most pronounced in the left auditory cortex. Importantly, listening to binary sounds alternating in pitch instead of duration did not yield group differences, confirming that the above results were driven by the hypothesized long-term 'duration' prior. These findings show that experience with a given language can shape a fundamental aspect of human perception – the neural processing of rhythmic sounds – and provides direct evidence for a long-term predictive coding system in the auditory cortex that uses auditory schemes learned over a lifetime to process incoming sound sequences.

*For correspondence: piermatteomorucci@gmail.com

Competing interest: The authors declare that no competing interests exist.

## eLife assessment

This study presents **important** observations about how the human brain uses long-term priors (acquired during our lifetime of listening) to make predictions about expected sounds - an open question in the field of predictive processing. The evidence presented is **solid** and based on state-of-the-art statistical analysis, but limited by a relatively low N and low magnitude for the interaction effect.

## Introduction

According to predictive coding theories of perception, sensory processes and perceptual decisions are described as a process of inference, which is strongly shaped by prior knowledge and predictions (*Clark, 2013*; *Friston, 2005*; *Rao and Ballard, 1999*). Predictions can be derived from different

sources of information, forming a hierarchical predictive system (*de Lange et al., 2018*). Each level of the predictive hierarchy houses an internal model encoding prior information about the structure of the external environment. When a prediction is violated, the prediction error is computed and used to adjust the corresponding prior and internal model. This results in a constantly evolving system that generates and refines predictions based on incoming sensory input and prior experience.

In the auditory domain, great progress in the understanding of the predictive capabilities of the auditory system has been made using the oddball design and its variations (see *Heilbron and Chait, 2018*, for a review). In these designs, participants are usually presented with sequences of tones encoding a certain rule that is then violated by a 'deviant' event. Such deviants elicit a sharp evoked response in the electroencephalography (EEG) signal which has been defined as 'mismatch negativity' (MMN). The MMN peaks at about 0.100–0.250 s from stimulus onset and exhibits enhanced intensity over secondary temporal, central, and frontal areas of topographic scalp maps (*Sams et al., 1985*; *Garrido et al., 2009*). Within the predictive coding framework, the MMN is putatively considered an index of cortical prediction error.

Functionally, the mechanism underlying the MMN operates over both conscious and preconscious memory representations. MMN responses to auditory violations are observable when the participant is not paying attention to the auditory task and have been reported even in states of sleep (*Sallinen et al., 1994*; *Sculthorpe et al., 2009*; *Strauss et al., 2015*) and coma (*Fischer et al., 2000*). Given its automatic nature, the anticipatory mechanism underlying the MMN has been suggested to reflect a form of '*primitive intelligence*' in the auditory cortex (*Näätänen et al., 2001*). In the present study, we show that life-long experience with a spoken language can shape this automatic anticipatory mechanism.

Experimental studies using the oddball design and its derivations have been important to unveil the sensitivity of the auditory predictive system to local statistical regularities and transition probabilities (*Heilbron and Chait, 2018*). However, these studies have primarily examined so-called contextual (or short-term) predictive signals. These predictions are usually based on rules acquired in the context of an experimental task – that is, rules linked to short-term memory – and have a short-lived impact on sensory processing. Yet, one core assumption of current predictive coding models is that the brain also deploys predictions based on long-term memory representations (*Seriès and Seitz, 2013*; *Yon et al., 2019*; *Teufel and Fletcher, 2020*). Such long-term predictions may emerge via learning of regularities and co-occurring patterns that are relatively stable throughout the lifespan of an organism. Because arising over long timescales, these experiential priors become encoded into the tuning properties of sensory cortices, forming a computational constraint on bottom–up sensory processing (*Teufel and Fletcher, 2020*).

Long-term priors may have long-lasting effects on perception. One example from the visual domain is the systematic bias in humans toward the perception of cardinal orientations (*Girshick et al., 2011*). This bias has been linked to the presence of a long-term prior that mirrors the statistics of the visual environment, that is, the preponderance of cardinal orientations in visual input (*Girshick et al., 2011*). Monkey studies on visual processing have shown that the visual system employs long-term priors to generate long-term predictions of incoming data (*Meyer and Olson, 2011*). Yet, whether similar predictive coding schemes subserve cortical computation in the human auditory system remains unsettled.

Here, we take a cross-linguistic approach to test whether the auditory system generates long-term predictions based on life-long exposure to auditory regularities, using rules that extend beyond those acquired in the recent past. Currently, one critical behavioral example of the effect of long-term experience on auditory perception is the influence of language on rhythmic grouping (*Iversen et al., 2008*; *Molnar et al., 2016*): Sequences of two tones alternating in duration are usually perceived by speakers of functor-initial languages (e.g., Spanish, English) as repetition of short–long groups separated by a pause, while speakers of functor-final languages (e.g., Basque, Japanese) report a bias for the opposite long–short grouping pattern. This perceptual effect has been linked to the co-occurrence statistics underlying the word-order properties of these languages. Specifically, the effect has been proposed to depend on the quasi-periodic alternation of short and long auditory events in the speech signal – reported in previous acoustic analyses (*Molnar et al., 2016*) – which reflect the linearization of function words (e.g., articles, prepositions) and content words (e.g., nouns, adjectives, verbs). In functor-initial languages, like English or Spanish, short events (i.e., function words; e.g., *un*,

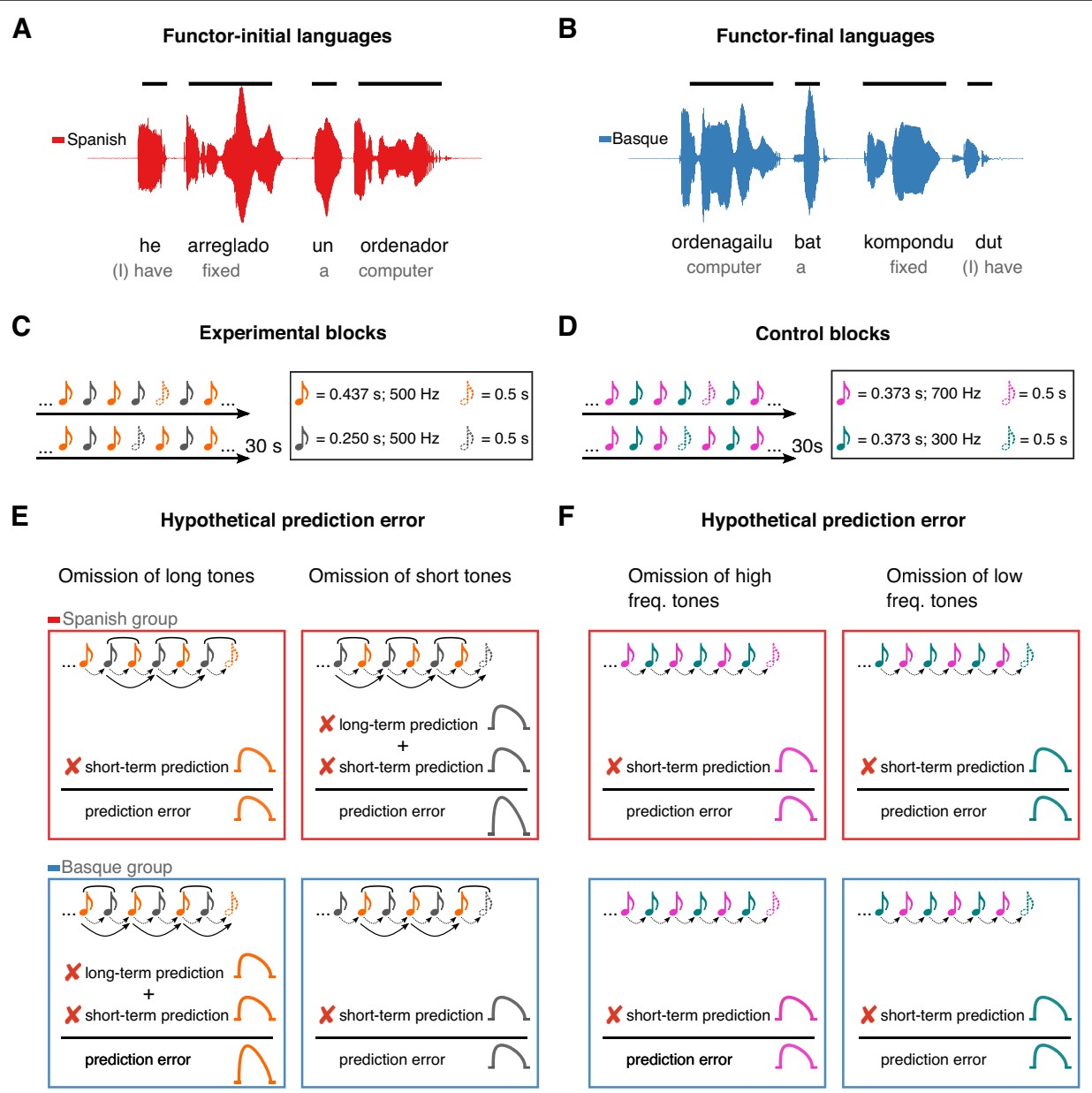

**Figure 1.** Experimental design and rationale of the study. Panels **A** and **B** illustrate the contrast between functor-initial and functor-final word order in Spanish and Basque, as well as its consequences on their prosodic structure. Panels **C** and **D** show the design of the experimental and control conditions, respectively. Notes represent individual tones. The structure of the design is the same in both conditions (*ababab*), with 30 s sequences of two tones alternating at fixed inter-stimulus intervals and occasional omissions. In the experimental condition, tones alternate in duration but not frequency, whereas in the control condition tones alternate in frequency but not duration. Panels **E** and **F** depict the hypothesized error responses associated with the different types of omissions for experimental and control conditions, respectively. Round brackets above the tones reflect the grouping bias of the two languages, based on their word-order constraints. Dotted lines reflect short-term predictions based on the transition probabilities of the previous stimuli. Solid lines reflect long-term predictions based on the phrasal chunking scheme of the two languages.

a) normally combine with long ones (i.e., content words; e.g., *ordenador*, computer) to form 'short–long' auditory chunks (*Figure 1A*). In contrast, in functor-final languages like Japanese and Basque, short events (i.e., function words; e.g., *bat*, a) normally follow long ones (i.e., content words; e.g., *ordenagailu*, computer), resulting in 'long–short' phrasal units (*Figure 1B*). Regular exposure to such language-specific phrasal structures has been proposed to underlie the automatic grouping biases of non-linguistic sounds (*Iversen et al., 2008*; *Molnar et al., 2016*), suggesting the presence of an auditory 'duration prior' that mirrors the word-order and prosodic properties of a given language.

We hypothesize that the auditory system uses the proposed 'duration prior' as an internal model to generate long-term predictions about incoming sound sequences. In predictive coding terms, our hypothesis posits that the human auditory system upweights neural activity toward the onset of certain high-level events, based on the statistics of a given language.

To test this hypothesis, two groups of Basque (*n* = 20) and Spanish (*n* = 20) dominant participants were presented with 30 s rhythmic sequences of two tones alternating in duration at fixed inter-stimulus intervals (*Figure 1C*), while magnetoencephalography (MEG) was monitoring their cortical activity. To measure prediction error, random omissions of long and short tones were introduced in each sequence. Omission responses allow to examine the presence of putative error signals decoupled from bottom–up sensory input, offering a critical test for predictive coding (*Walsh et al., 2020*; *Heilbron and Chait, 2018*).

If, in line with our hypothesis, the human auditory system uses long-term linguistic priors as an internal model to predict incoming sounds, the following predictions ensue. The omission of a long tone should represent the violation of two predictions in the Basque, but not in the Spanish group: a short-term prediction based on the statistics of the previous stimuli (i.e., a prediction about a new tone), and a long-term prediction based on the statistics of the Basque's phrasal structure (i.e., a prediction about a new phrasal chunk). Consequently, such an omission response should lead to a larger prediction error in the Basque compared to the Spanish group (*Figure 1E*). An orthogonally opposite pattern is expected when the deviant event is reflected in the omission of a short tone (*Figure 1E*).

The expectation that stronger error responses would be elicited by the omission of the first element rather than the second element of a perceptual chunk ('long' for the Basque, 'short' for the Spanish group) is primarily based on previous work on rhythm and music perception (e.g., *Ladinig et al., 2009*; *Bouwer et al., 2016*; *Brochard et al., 2003*; *Potter et al., 2009*). These studies have shown that the amplitude of evoked responses is larger when deviants occur at the 'start' of a perceptual group and decline toward the end of the chunk, suggesting that the auditory system generates predictions about the onset of higher-level, internally formed auditory chunks.

We tested the predictions above against a control condition having the same alternation design as the experimental condition, but with the two tones alternating in pitch instead of duration (*Figure 1D*). Here, no difference between groups is expected, as both groups should rely on short-term, but not long-term priors (*Figure 1F*). Finally, we performed reconstruction of cortical sources to identify the regions supporting long-term auditory priors.

## Results

We first examined the responses evoked by the omissions of tones. MEG responses time-locked to the onset of the tones and omissions from the Basque and Spanish dominant groups were pulled together and compared via cluster-based permutation test analysis (*Maris and Oostenveld, 2007*). Cluster analysis was performed over several time points to identify spatiotemporal clusters of neighboring sensors where the two conditions differ (*Methods*). This analysis revealed an early effect of omission responses arising around 0.100 s from deviance onset (p < 0.0001), including several channels over the entire scalp (*Figure 2A, B*). The latency and topographical distribution of the effect resemble the one elicited by a classical mismatch response, with strong activations over left and right temporal regions (*Figure 2A, B*). A smaller cluster (p = 0.043) with lower amplitude was also detected in an earlier time window (~0.030–0.050 s post-stimulus/omission onset). This cluster primarily includes left centro-temporal channels. The directionality of the effect is the same as the later cluster, that is, larger responses to omissions compared to tones (*Figure 2—figure supplement 1*). This finding aligned with previous reports showing that the omission of an expected tone in a regular sequence of sounds generates larger event-related fields (ERFs) than an actual tone (e.g., *Yabe et al., 1997*; *Raij et al., 1997*).

Our main question was on the presence of long-term predictions induced by the linguistic background of the participants during the processing of simple binary auditory sequences. To assess this, we tested for the presence of an interaction effect between the linguistic background of the participants (Basque, Spanish) and the type of tone omitted (long, short) in modulating the amplitude of the MMN. Omission MMN responses in this analysis were calculated by subtracting the ERF elicited by a given type of tone (i.e., long, short) from its corresponding omission (*Garrido et al., 2009*). A

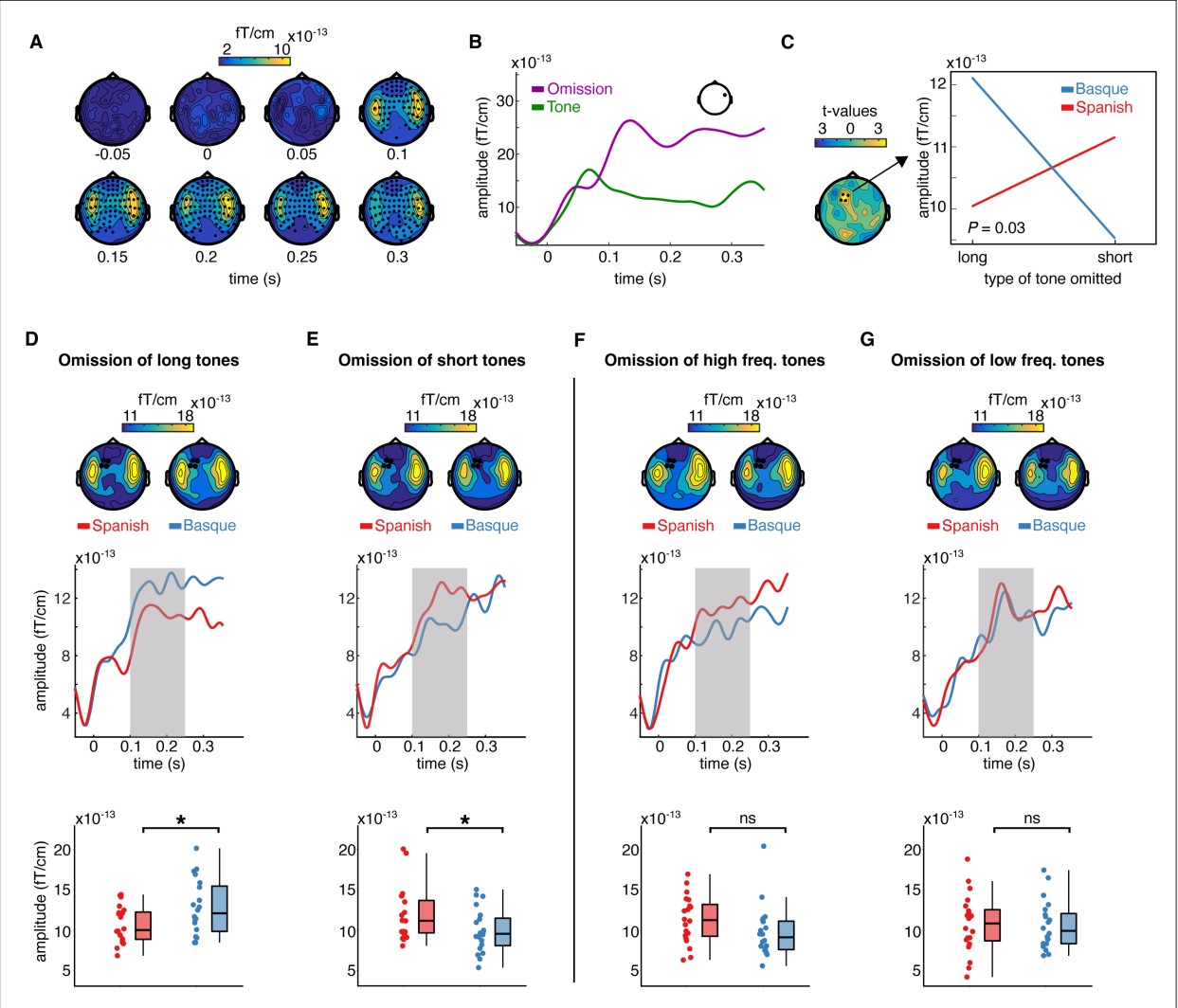

**Figure 2.** Sensor-level topography and time course of neural responses to omitted sounds across groups and conditions. (**A**) shows the temporal unfolding and topographical distribution of the overall effect of omission (omissions minus tones). Channels belonging to the significant cluster are highlighted. (**B**) shows the event-related field (ERF) generated by omissions and tones in a representative channel. (**C**) (left) shows the topography of the *t* distribution of the interaction effect between the language background of the participants (Spanish, Basque) and the type of omission mismatch negativity (MMN) (short, long). Channels belonging to the significant interaction cluster are highlighted. The interaction effect was present only in the experimental condition. Panel C (right) shows the averaged magnetoencephalography (MEG) activity over the 0.100–0.250 s time window and channels belonging to the significant cluster for each group and condition separately. (**D–G**) show the effect of language experience in modulating the amplitude of the omission MMN associated with each experimental and control contrast. Topographies (top) show the scalp distribution of the averaged activity over the 0.100–0.250 s time window. Channels belonging to the significant interaction cluster are highlighted. ERFs (middle) show the temporal unfolding of brain activity averaged over the channels belonging to the significant interaction cluster for each contrast and group. The shaded area indicates the time window of interest for the statistical analysis. Boxplots (down) show the mean MEG activity for each participant over the 0.100–0.250 s time window and the channels belonging to the significant interaction cluster. The center of the boxplot indicates the median, and the limits of the box define the interquartile range (IQR = middle 50% of the data). The notches indicate the 95% confidence interval around the median. Dots reflect individual subjects (n = 20 per group). In D–G, asterisks indicate statistical significance for each contrast using a one-sided, independent sample *t*-test with false discovery rate (FDR) correction for multiple comparisons (statistical significance: * signifies p < 0.05, *ns* signifies p > 0.05).

The online version of this article includes the following figure supplement(s) for figure 2:

**Figure supplement 1.** Early neural response to omitted sounds.

**Figure supplement 2.** Effects of omission type and language background.

cluster-based permutation test was used to test the interaction effect between omission type (long vs short) and the linguistic background of the participants (Basque vs Spanish). As the cluster-based permutation test is designed to compare two conditions at a time, we tested for an interaction effect by subtracting the MMN elicited by the omission of a long tone from the MMN elicited by the omission of a short tone for each participant, and then compared the resulting differences between groups.

For this and the following contrasts, we selected a predefined time window of interest between 0.100 and 0.250 s, which covers the typical latency of the MMN (*Näätänen et al., 2007*; *Garrido et al., 2009*). Cluster analysis revealed a significant interaction (p = 0.03), which was particularly pronounced over left frontotemporal channels (see *Figure 2C*). To unpack the interaction, we averaged data samples for each participant and condition over the channels belonging to the significant cluster and time points of interest. This resulted in two ERFs for each participant, one for each type of omission MMN (long, short). We then compared ERFs for each omission type between the two groups using a one-sided independent sample *t*-test, testing the hypothesis that participants deploy long-term expectations about the onset of abstract language-like grouping units (*Figure 1E*). Specifically, we compared (1) the omission MMN responses generated by the omission of long tones in the Basque vs the Spanish group, and (2) the MMN responses generated by the omission of short tones in the Basque vs the Spanish group. Consistent with the hypothesis, we found that omissions of long tones generated a larger MMN response in the Basque compared Spanish group ($t(38) = 2.22$; p = 0.03 FDR-corrected; $d = 0.70$), while the omission of short tones generated a larger omission MMN in the Spanish compared to Basque group ($t(38) = -2$; p = 0.03 FDR-corrected; $d = 0.63$) (*Figure 2D, E*). Notice that, given the structure of our design, a *between-group* comparison (e.g., comparing the ERF between the Basque and Spanish groups) is more suited to test our hypothesis than a *within-group* comparison (e.g., comparing the ERF evoked by the omission of long vs short tones within the Basque group), as the pre-stimulus baseline activity is virtually identical across conditions only in the *between-group* contrast.

To further assess that the interaction was driven by the hypothesized long-term 'duration prior', the same analysis pipeline was applied to the data from the control condition. Here, no significant omission type × language background interaction was detected (no cluster with p < 0.05). To further check that no interaction was present in the control study, we averaged the data samples over the channels and time points in which we detected a significant interaction in the test condition and ran an independent sample *t*-test by comparing MMN responses elicited by the omission of high- and low-frequency tones in both groups (*Methods*). Even within this subset of channels, no between-group difference was detected between MMN responses evoked by omissions of high- ($t(38) = -1.6$; p = 0.12 FDR-corrected; $d = -0.51$), and low-frequency tones ($t(38) = -1.1$; p = 0.55 FDR-corrected; $d = -0.04$) (*Figure 2F, G*).

A linearly constrained minimum variance (LCMV) beamformer approach (*Van Veen et al., 1997*) was used to reconstruct the cortical sources of the MEG signal. We first focused on the source activity underlying the effect of omission (*Figure 2A, B*). Source activity was calculated for the epochs averaged in the 0.100–0.250 s interval for both tones and omissions. In order to isolate regions underlying the effect of omission, whole-brain maps for the ratio of the source activity associated with omission responses and tones were created (*Figure 3A*). A large network of regions showed a stronger response to omissions compared to auditory tones, including the bilateral inferior frontal gyri, premotor cortices, angular gyri, as well as the superior temporal gyri (*Figure 3A*).

We then examined the cortical origin of the interaction effect that emerged at the sensor level (*Figure 2C*). In line with the MMN analysis at the sensor level, we focused on the source activity of the difference between omissions and tones for each omission type (short, long) and language group (Basque, Spanish). This analysis was aimed at identifying the cortical origin of the hypothesized long-term predictions. We hypothesized that such long-term priors might be linked to long-term experience with the rhythmic properties of the two languages. As such, they would be expected to arise around early auditory areas, such as the superior temporal gyrus (STG). An alternative hypothesis is that these priors are linked to the abstract syntactic structure of the two languages. Under this account, long-term predictions would be generated via long-range feedback from regions associated with syntactic processing, such as the left inferior frontal gyrus (IFG) (*Ben Shachar et al., 2003*). To disentangle these possibilities, we performed an analysis on three regions of interest (ROIs, based on the Brainnetome atlas, *Fan et al., 2016*) within the left STG, (Brodmann areas (BAs) 41/42 of the

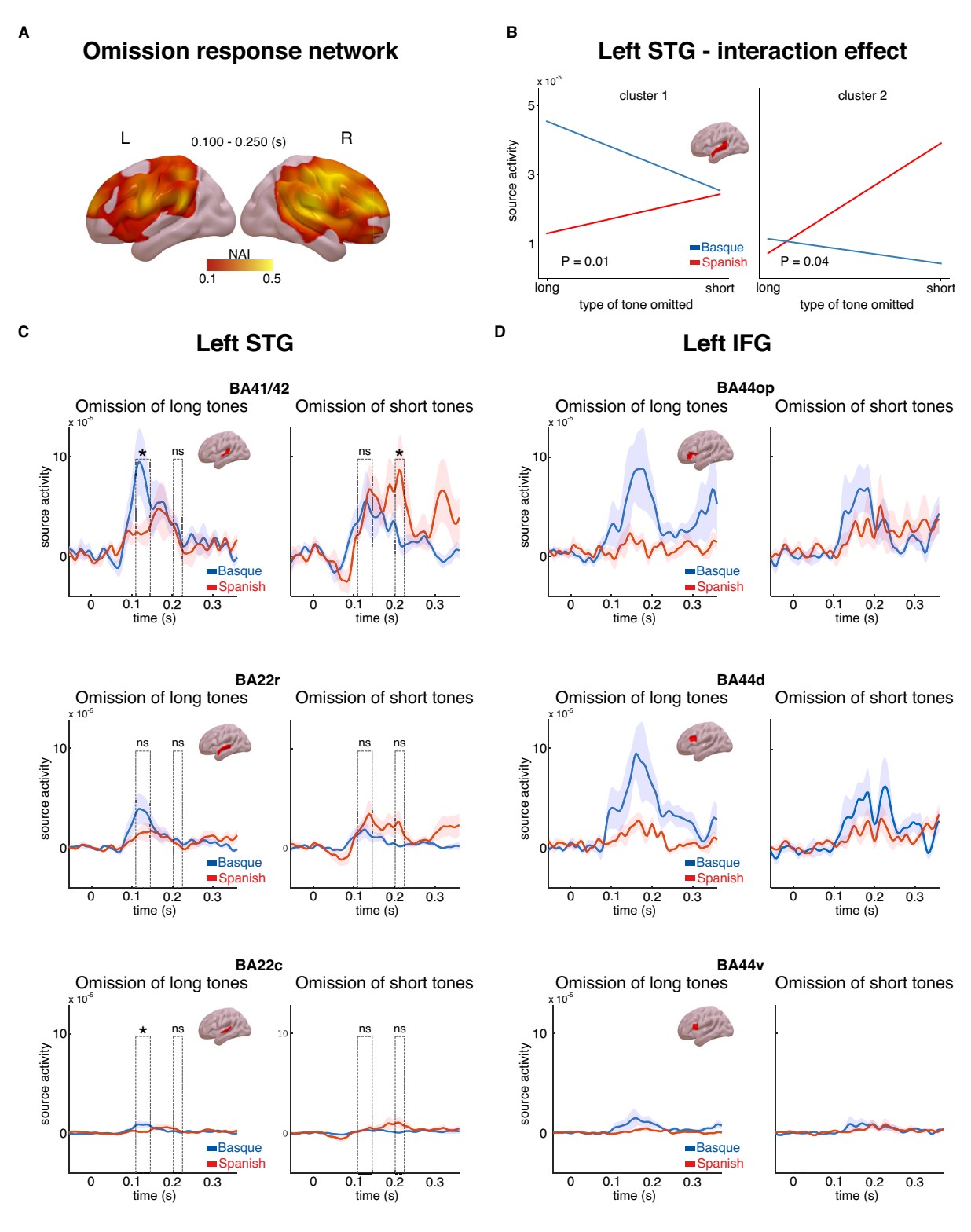

**Figure 3.** Source activity underlying the omission response network and long-term predictions. Panel **A** shows brain maps representing the ratio of source activity of tone to the omission (neural activity index, NAI = $S_{Omission}/S_{Tone}$) over the 0.100–0.250 s time window. Panel **B** shows the source activity peaks from the two clusters in the left superior temporal gyrus (STG) for each group and condition separately. Panels **C** and **D** show the time course of source activity associated with the omission mismatch negativity (MMN) over distinct regions of interest (ROIs) of the left STG and inferior frontal gyrus (IFG). Dashed rectangles indicate the two temporal clusters within the 0.100–0.250 s time window. Error bars reflect standard errors (n = 20 per group).

*Figure 3 continued on next page*

*Figure 3 continued*

In Panels C and D, asterisks indicate statistical significance for each contrast using a one-sided, independent sample *t*-test (statistical significance: * signifies p < 0.05, *ns* signifies p > 0.05).

auditory cortex, rostral portion of BA 22, caudal portion of BA 22), and three ROIs within the left IFG (dorsal, ventral, and opercular portions of BA 44). We restricted our analysis to the left hemisphere only, which is where the significant interaction effect emerged at the sensor level (*Figure 2C*). We first performed a cluster-based permutation analysis in the whole left STG and the left IFG in the 0.100–0.250 s time interval, testing for the presence of an interaction between omission type and language group. This strategy is similar to the one performed at the sensor level, but is more time-sensitive as individual time points in the 0.100–0.250 s time interval are considered. No significant interaction was detected in the left IFG (no cluster with p < 0.05), while two temporally distinct effects emerged in the left STG: an early effect arising in the 0.110–0.145 s time interval (p = 0.01) and a later effect in the 0.200–0.220 s (p = 0.04) (*Figure 3B–D*). It is possible that these two clusters reflect different temporal responses of the two groups to long and short omissions. To better understand the nature of the interaction effect within the left STG, pairwise comparisons were performed on each ROI and cluster using one-sided independent sample *t*-tests, following the same contrasts that were performed at the sensor level. Despite the high redundancy of activity across neighboring brain regions due to MEG source-reconstruction limitations (*Bourguignon et al., 2018*), we were interested in verifying whether the more robust interaction effect, both in magnitude and reliability, was emerging in primary (BA 41/42) or associative (BAs 22) auditory regions. Given the limitations of the cluster-based statistical approach in the definition of the timing of significant effects (*Sassenhagen and Draschkow, 2019*), peak activity within each temporal cluster in the STG was selected and used for the pairwise comparisons on each ROI (early peak: 0.120 s; late peak: 0.210 s). Pairwise comparisons showed that the omission of long tones generated larger responses on the latency of the early peak (0.120 s) in the Basque compared to the Spanish dominant group over BA 41/42 and BA 22c BA 41/42 ($t(38) = 1.88$, p = 0.03, $d = 0.59$); BA 22c ($t(38) = 2.04$, p = 0.02, $d = 0.64$), while no difference emerged for later peak responses (0.210 s) (BA 41/42: $t(38) = 0.59$, p = 0.27, $d = 0.32$; BA 22c: $t(38) = -1.14$, p = 0.87, $d = -0.36$) (*Figure 3C*). On the contrary, the omission of short tones led to stronger responses over the later peak latency (0.210 s) in the Spanish compared to the Basque group in BA 41/42 ($t(38) = -1.98$, p = 0.02, $d = -0.62$), while no difference was observed on the latency of the earlier peak (BA 41/42: $t(38) = 0.32$, p = 0.62, $d = 0.10$) (*Figure 3C*). No significant effect emerged in BA22r. Overall, these findings suggest that segments of the left STG, in particular BA41/42, exhibit distinct sensitivity to omission responses in the two groups. The Basque group shows larger responses to the omission of long tones at an earlier time interval, whereas the Spanish group displays increased responses to the omission of short tones at a later time interval (*Figure 3C*).

Besides documenting an interaction, which was the analysis of interest of our study, we also searched for a main effect of omission type. A cluster-based permutation test was used to compare the MMN responses elicited by omissions of long tones and omissions of short tones averaged across the two groups. The results showed that long-tone omissions generated a larger omission MMN response than short-tone omissions (p = 0.03), with the cluster including several frontal channels (*Figure 2—figure supplement 2*). This effect was consistent in the Basque (p = 0.003), but not in the Spanish group (no clusters with p < 0.05). No main effect of language background was detected (no clusters with p < 0.05) (*Figure 2—figure supplement 2*). In the control condition, neither a main effect of omission type nor an effect of language background was detected (no clusters with p < 0.05).

## Discussion

By comparing MEG data from native speakers of functor-initial (i.e., Spanish) and functor-final languages (i.e., Basque) listening to simple binary sequences of tones with occasional violations, we show that experience with a given language can shape a very simple aspect of human perception, such as the neural processing of a binary rhythmic sound. This finding suggests that the human auditory system uses structural patterns of their native language to generate predictive models of non-linguistic sound sequences. This result highlights the presence of an active predictive system that relies on natural sound statistics learned over a lifetime to process incoming auditory input.

We first looked at the responses generated by sound omissions. In line with previous reports, we found that omissions of expected tones in an auditory sequence generate a sharp response in the ERF. Such omission responses have been suggested to reflect pure error signals decoupled from bottom–up sensory input (*Hughes et al., 2001*; *Wacongne et al., 2011*). On the other hand, other studies have proposed that omission responses could reflect pure predictions (*Bendixen et al., 2009*; *SanMiguel et al., 2013*). While the exact nature of such responses is currently debated and likely dependent on factors such as task and relevance, the latency and topography of the omission response in our data resemble those evoked by a classical mismatch response (*Figure 2A, B*). Analysis of cortical sources also supports this interpretation. Indeed, source activity associated with omissions leads to stronger responses compared to tones over a distributed network of regions, including the bilateral inferior frontal gyri, premotor areas, angular gyri, and right STG. Since sensory predictive signals primarily arise in the same regions as the actual input, the activation of a broader network of regions in omission responses compared to tones suggests that omission responses reflect, at least in part, prediction error signals.

Importantly, we showed that when an unexpected omission disrupts a binary sequence of sounds, the amplitude of the omission MMN varies orthogonally depending on the speaker's linguistic background. Omissions of long auditory events generate a larger omission MMN in the Basque compared to the Spanish group, while omissions of short sounds lead to a larger omission MMN responses in the Spanish compared to the Basque group. We hypothesized that this effect is linked to a long-term 'duration prior' originating from the acoustic properties of the two languages, specifically from the alternation of short and long auditory events in their prosody. Importantly, no difference between groups was detected in a control task in which tones alternate in frequency instead of duration, suggesting that the reported effect was driven by the hypothesized long-term linguistic priors instead of uncontrolled group differences.

It is important to note that both Spanish and Basque speakers are part of the same cultural community in Northern Spain. These languages share almost the same phonology and orthography. However, Basque is a non-Indo-European language (an isolated language) with no typological relationship with Spanish. It is thus very unlikely that the current findings are driven by cultural factors that are not language-specific (e.g., exposure to different musical traditions or educational and writing systems).

How would such long-term priors arise? One possible interpretation is that long-term statistical learning of the duration prosodic pattern of native language shapes the tuning properties of early auditory regions, affecting predictive coding at early stages. Such language-driven tuning is arguably important for reducing the prediction error during the segmentation of speech material into phrasal units, as it allows the auditory system to generate a functional coding scheme, or auditory template, against which the incoming speech input can be parsed. Such an auditory template is likely to be recycled by the auditory system to build top–down predictive models of non-linguistic auditory sequences.

The idea that the auditory system implements long-term predictions based on the prosodic structure of the native language could explain the previously reported behavioral influence of language experience on rhythmic grouping (*Iversen et al., 2008*; *Molnar et al., 2016*): when listening to sequences of two tones alternating in duration, like those used in the present study, speakers of functor-initial languages report to perceive the rhythmic sequences as a repetition of 'short–long' units, while speakers of functor-final languages have the opposite 'long–short' grouping bias (*Iversen et al., 2008*; *Molnar et al., 2016*). Despite lacking a direct behavioral assessment (but see *Molnar et al., 2016*, for related behavioral evidence), our results indicate that this perceptual grouping effect can be explained within a predictive coding framework that incorporates long-term prior knowledge into perceptual decisions. Under such an account, the auditory system internalizes the statistics underlying the prosodic structure of language and uses this knowledge to make long-term predictions of incoming sound sequences. Such long-term predictions would bias auditory processing at early, rather than later decision-making stages, affecting how rhythmic sounds are experienced.

Our work capitalized on a specific aspect of natural sound acoustic – the duration pattern in Basque and Spanish prosody – as a testbed to assess the presence of long-term priors in the auditory system. Despite our work being restricted to this specific feature, it is likely that the auditory system forms several other types of long-term priors using the spectrotemporal features that dominate the auditory environment. Support for this claim comes from (1) studies showing that the human auditory system uses the statistics underlying the acoustic structure of speech and music to form perceptual grouping

decisions (*Młynarski and McDermott, 2019*); and (2) behavioral experiments reporting off-line effects of language experience on auditory perception based on different acoustic features (*Liu et al., 2023*). For instance, native speakers of languages in which pitch carries phonemically meaningful information (i.e., tone languages, e.g., Mandarin Chinese) benefit from a behavioral advantage in non-linguistic pitch discrimination tasks as compared to speakers of non-tone languages like English (*Bidelman et al., 2013*). Similarly, speakers of languages that use duration to differentiate between phonemes (e.g., Finnish, Japanese) manifest an enhanced ability to discriminate the duration of non-linguistic sounds (*Tervaniemi et al., 2006*). Our results, in conjunction with these studies, suggest that the auditory system forms long-term priors and predictions over development, using the co-occurrences that dominate the natural stimulus statistics. Yet, our results leave open the question of whether these long-term priors can be updated during adulthood, following extensive exposure to new statistical dependencies. This can be tested by exposing adult speakers to natural sounds encoding rules that 'violate' the long-term prior (e.g., a language with opposite prosodic structure) and exploring the effects of such short-term exposure to behavioral and neural performance.

One potential alternative to the conjecture that the 'duration prior' is linked to the spectro-temporal features of a language is that the prior depends on abstract syntactic/word-order rules. This latter account would predict that violations of long-term predictions in our study would lead to larger error responses in regions sensitive to syntactic variables, such as the left IFG (*Ben Shachar et al., 2003*). Instead, the former account would predict that violations of long-term predictions elicit stronger responses in early left-lateralized auditory regions, which are putatively associated with early speech processing (*Bhaya-Grossman and Chang, 2022*). The reconstruction of cortical sources associated with the omission of short and long tones in the two groups showed that an interaction effect mirroring the one at the sensor level was present in the left STG, but not in the left IFG (*Figure 3B–D*). Pairwise comparisons within different ROIs of the left STG indicated that the interaction effect was stronger over primary (BA 41/42) rather than associative (BAs 22) portions of the auditory cortex. Overall, these results suggest that the 'duration prior' is linked to the acoustic properties of a given language rather than its syntactic configurations.

Our results are in line with predictive coding models stating that predictions are organized hierarchically. When two predictive signals, one short-term and one long-term are violated, the amplitude of the prediction error is larger compared to a scenario in which only one short-term prediction is violated. This result complements previous studies using the local–global design showing that the same deviancy presented in different contexts gives rise to different error signals, such as the MMN and the P3 (*Bekinschtein et al., 2009*; *Wacongne et al., 2011*). These studies provide empirical evidence that predictive coding of auditory sequences is organized at different functional levels, with early sensory regions using transition probabilities to generate expectations about the present, and frontal and associative regions inferring the global structure of an auditory event. Our results extend this work by providing direct evidence for the presence of a system in the auditory cortex that uses long-term natural sound statistics to generate long-term predictions. This interpretation is also supported by the reconstruction of cortical sources. Indeed, while the overall omission effect is larger in the right hemisphere (*Figures 2A and 3A*), the interaction effect arises in the left hemisphere (*Figures 2C and 3B, C*). This finding further suggests that distinct cortical systems, supporting different predictive models, underlie the generation of the omission MMN.

Our findings are also consistent with more recent predictive coding models incorporating the idea of 'stubborn' predictive signals – that is, predictions resilient to model updates. Unlike short-term expectations, long-term predictions are usually implemented as a computational constraint on input data, thus being largely unaffected by short-term experience (*Teufel and Fletcher, 2020*). In our study, the deployment of long-term predictions does not represent an effective coding strategy to perform the task. Yet, listeners still seem to assign different weights to incoming data, using a 'default' predictive coding scheme that resembles the segmentation strategy used to parse speech material. Why should a neural system rely on such stubborn priors even when irrelevant to solving a given perceptual task? One possibility is that implementing stable priors as a constraint on perception is computationally less expensive in terms of metabolic costs than recalibrating cortical internal models anew based on any type of novel experience. Another possibility is that relying on unchanging predictive schemes helps the system to form coherent models in front of environmental contingencies, thus reflecting an effective computational strategy for the reduction of the long-term prediction error. Defining how

stubborn predictions emerge during learning and what their computational role is represents an important challenge to understanding the role of prior experience in perceptual inference.

We also reported a main effect of omission type, indicating that the MMN generated by the omission of a long tone was generally larger compared to that generated by the omission of a short one. Because such group effect was consistent only in the Basque group, it is possible that it merely reflects a larger sensitivity of the auditory system of this group to the omission of long events, in line with the interaction reported above. Alternatively, this effect could be driven by the fact that, during language processing, major predictive resources are invested in predicting the onset of long events, compared to short ones, as the formers usually refer to content words that is, semantically relevant events. Consequently, the auditory system may apply a similar predictive scheme also during the processing of non-linguistic sound sequences, independently of language background. Independently on the interpretation, the lack of a main effect of omission type in the control condition suggests that the long omission effect is driven by experience with the native language.

Our results also refine previous studies showing modulatory effects of (long-term) musical expertise on the MMN (e.g., *Vuust et al., 2005*; *Vuust et al., 2009*). These studies indicate that responses to violation during auditory rhythm perception are larger when the listener is an expert musician compared to a non-musician, pointing to the role of long-term auditory experience in shaping early predictive mechanisms. In our study, we manipulated long-term prediction orthogonally, with clear-cut predictions about the effect of language experience on early auditory predictive processing. Our results thus provide direct evidence for the presence of an active system in the auditory cortex that uses long-term priors to constrain information processing of incoming auditory stimuli.

## Materials and methods

### Participants

In total, 20 native speakers of Spanish (mean age: 25.6 years, range: 20–33, 13 females) and 20 native speakers of Basque (mean age: 27.11 years, range: 22–40, 17 females) took part in the experiment. It must be noted that in the original *Molnar et al., 2016* experiment, a sample size of 16 subjects per group was sufficient to detect a behavioral perceptual grouping effect (under the request of a reviewer, we report a post hoc power analysis indicating an achieved power of 46% for medium effect sizes ($d$ = 0.5, and alpha = 0.05, one-sided test) in a between-groups design with 20 subjects per group; while a sensitivity analysis indicates that the experiment possesses 80% power for effect sizes of $d$ = 0.8 and above). Members of the two groups were selected based on self-reported scores for exposure (percentage of time exposed to a given language at the time of testing) and speaking (percentage of time speaking a given language at the time of testing). Participants from the Basque group were living in a Basque-speaking region of the Basque Country. They all reported having learned Basque as a first language, being primarily exposed to Basque during daily life (mean exposure: 69%; standard deviation [SD]: 13.28; range: 50–90%) and using it as a main language for communication (mean speaking: 77%; SD: 10.56; range: 60–90%). All native speakers of Basque reported having learned Spanish as a second language. However, they had overall low exposure (mean exposure: 22%; SD: 10.31; range: 10–40%) and speaking scores for Spanish (mean speaking: 17%; SD: 7.33; range: 10–30%). In this respect, it is important to notice that previous behavioral studies on perceptual grouping in Basque bilinguals showed that language dominance is the main factor driving non-linguistic rhythmic grouping (*Molnar et al., 2016*). Therefore, despite limited exposure to the Spanish language, the formation of the hypothesized 'duration prior' in the Basque group should be primarily linked to experience with the dominant language (i.e., Basque), with no or only minimal influence from Spanish. Participants from the Spanish dominant group were coming from different regions of Spain. All of them learned Spanish as their first language, and had high self-reported scores for Spanish exposure (mean exposure: 79%; SD: 9.67; range: 60–100%) and speaking (mean speaking: 88.5%; SD: 6.7; range: 80–100%). Spanish participants reported having learned a second or third language after childhood (e.g., Basque, English, Italian, and Catalan).

Participants were recruited through the participant recruitment system of the Basque Center on Cognition, Brain and Language. The experiment and methods received approval from both the ethical committee and scientific committee of the Basque Center on Cognition, Brain and Language (Ethics approval number: 18072018M), following with the principles of the Declaration of Helsinki.

Written informed consent was obtained from all participants in line with the guidelines of the Research Committees of the BCBL.

## Stimuli and experimental design

Stimuli were created using Matlab Psychtoolbox and presented binaurally via MEG-compatible headphones. Experimental stimuli consisted of 60 sequences of two tones alternating in duration (short tones: 0.250 s; long tones: 0.437 s) with fixed inter-stimulus intervals (0.020 s). Both long and short tones had a frequency of 500 Hz. The beginning and end of each tone were faded in and out of 0.015 s. Overall, each sequence consisted of 40 short- to long-tone pairs, for a total of 80 unique tones per sequence, and lasted around 30 s. Half of the sequences started with a long tone and half with a short tone. The beginning and the end of each sequence were faded in and faded out of 2.5 s to mask possible grouping biases. In each sequence, two to six tones were omitted and substituted with a 0.6-s silence gap. The larger gap was introduced to avoid that activity related to the onset of the tone following the omission overlaps with the activity generated by the omitted tone. Tone omissions occurred pseudorandomly, for a total of 240 omissions (120 short and 120 long). The pseudo-randomization of the omissions consisted in separating the omissions within each sequence of at least seven tones. In the control condition, sequences consisted of tones alternating in frequency at fixed inter-stimulus intervals (0.020 s). High-frequency tones had a frequency of 700 Hz, while low-frequency tones had a frequency of 300 Hz. Both high- and low-frequency tones had an overall duration of 0.343 s. This duration was selected to keep the overall length of the sequences equal to that of the test condition, by keeping the total number of 80 tones per sequence. As in the test condition, tones and sequences were faded in and out of 0.015 and 2.5 s, respectively. In each sequence, two to six tones were omitted and substituted with a 0.600-s silence gap.

Overall, the experiment was divided into two main blocks: test and control. The order in which the blocks were presented was counterbalanced across participants. Each block consisted of 60 sequences and lasted around 35 min. Each sequence was separated by an 8-s silence gap. Every 20 sequences, a short pause was introduced. The end of each block was followed by a longer pause.

Participants were requested to minimize movement throughout the experiment, except during pauses. Subjects were asked to keep their eyes open, to avoid eye movements by fixating on a cross on the screen. Similar to previous studies, the only task that was asked to subjects was to count how many omissions were present in each sequence (e.g., *Bekinschtein et al., 2009*) – and report it at the end of the sequence during the 8-s silence gap. Participants only received instructions at the very beginning of the task, and no verbal or written instructions were introduced during the task.

## MEG recordings

Measurements were carried out with the Elekta Neuromag VectorView system (Elekta Neuromag) of the Basque Center on Cognition Brain and Language, which comprises 204 planar gradiometers and 102 magnetometers in a helmet-shaped array. Electrocardiogram (ECG) and electrooculogram (EOG) (horizontal and vertical) were recorded simultaneously as auxiliary channels. MEG and auxiliary channels were low-pass filtered at 330 Hz, high-pass filtered at 0.03 Hz, and sampled at 1 kHz. The head position with respect to the sensor array was determined by five head-position indicator coils attached to the scalp. The locations of the coils were digitized with respect to three anatomical landmarks (nasion and preauricular points) with a 3D digitizer (Polhemus Isotrak system). Then, the head position with respect to the device origin was acquired before each block.

## Preprocessing

Signal space separation correction, head movement compensation, and bad channels correction were applied using the MaxFilter Software 2.2 (Elekta Neuromag). After that, data were analyzed using the FieldTrip toolbox (*Oostenveld et al., 2011*) in Matlab (MathWorks). Trials were initially epoched from 1.200 s before to 1.200 s after the onset of each tone or omitted tone. Epochs time-locked to the onset of short and long tones were undersampled to match approximately the number of their corresponding omissions. Trials containing muscle artifacts and jumps in the MEG signal were detected using an automatic procedure and removed after visual inspection. Subsequently, independent component analysis (*Makeig et al., 1995*) was performed to partially remove artifacts attributable to eye blinks and heartbeat artifacts (*Jung et al., 2000*). To facilitate the detection of components

reflecting eye blinks and heartbeat artifacts, the coherence between all components and the ECG/EOG electrodes was computed. Components were inspected visually before rejection. On average, we removed 14.28% (SD = 5.71) of the trials and 2.45 (SD = 0.55) components per subject. After artifact rejection, trials were low-pass filtered at 40 Hz and averaged per condition and per subject. ERFs were baseline corrected using the 0.050 s preceding trial onset and resampled to 256 Hz. The latitudinal and longitudinal gradiometers were combined by computing the root mean square of the signals at each sensor position to facilitate the interpretation of the sensor-level data.

## ERF analysis

Statistical analyses were performed using FieldTrip (*Oostenveld et al., 2011*) in Matlab 2014 (MathWorks) and R studio for post hoc analysis. For data visualization, we used Matlab or FieldTrip plotting functions, R studio and the RainCloud plots tool (*Allen et al., 2019*). Plots were then arranged as cohesive images using Inkscape (https://inkscape.org/). All comparisons were performed on combined gradiometer data. For statistical analyses, we used a univariate approach in combination with cluster-based permutations (*Maris and Oostenveld, 2007*) for family-wise error correction. This type of test controls the type I error rate in the context of multiple comparisons by identifying clusters of significant differences over space and time, instead of performing a separate test on each sensor and sample pair. Two-sided paired- and independent-samples *t*-tests were used for within- and between-subjects contrasts, respectively. The minimum number of neighboring channels required for a sample to be included in the clustering algorithm was set at 3. The cluster-forming alpha level was set at 0.05. The cluster-level statistic was the maximum sum of *t*-values (maxsum) and the number of permutations was set to 100,000. To control for the false alarm rate, we selected the standard $\alpha$ = 0.05. For the first analysis only, in which we compared ERF generated by pure tones vs omitted tones, we used a time window between −0.050 and 0.350 s and considered both the spatial and temporal dimensions in the cluster-based permutation test. This explorative analysis was performed to assess the effect of unexpected omission, as well as its temporal unfolding. In all the remaining analyses, MMN responses were calculated by subtracting the ERFs of each type of tone (i.e., long, short) from the ERFs of the corresponding omission. Moreover, all the contrasts were conducted using the average activity in the latency range between 0.100 and 0.250 s, which covers the typical latency of the MMN (*Näätänen et al., 2007*; *Garrido et al., 2009*). This approach uses spatial clusters to test for the difference between conditions. When multiple clusters emerged from a comparison, only the most significant cluster was reported. When an interaction effect was detected, post hoc analyses were performed to assess its directionality. This was done by averaging data for each participant and condition over the preselected latency and channels belonging to the significant clusters (see *Results*). The ERFs generated by this averaging were compared using one-sided independent sample *t*-tests. All p-values resulting from the post hoc *t*-test comparisons were FDR-corrected for multiple comparisons. All effect sizes reported are Cohen's *d* (*Cohen, 2013*).

## Source reconstruction

Source reconstruction mainly focused on the statistically significant effects observed at the sensor-level ERF analysis. Individual T1-weighted magnetic resonance imaging (MRI) images were first segmented into scalp, skull, and brain components using the segmentation algorithm implemented in Fieldtrip (*Oostenveld et al., 2011*). A standard Montreal Neurological Institute (MNI) brain template available in SPM toolbox was used for the participants whose MRI was not available. Co-registration of the anatomical MRI images with MEG signal was performed using the manual co-registration tool available in Fieldtrip. The source space was defined as a regular 3D grid with a 5-mm resolution, and the lead fields were computed using a single-sphere head model for three orthogonal source orientations. The whole-brain cortical sources of the MEG signals were estimated using an LCMV beamformer approach (*Van Veen et al., 1997*). Only planar gradiometers were used for modeling source activity. First, we compared the average source activity associated with pure omissions and standard tones in the time range of 0.100–0.250 s after the stimulus onset, extracted from the individual epochs. The covariance matrix used to derive LCMV beamformer weights was calculated from the 0.2 s preceding and 0.4 s following the onset of events (i.e., tones and omissions). Neural activity index (NAI) was computed as the ratio of source activity of tone to the omission (NAI = $S_{Omission}/S_{Tone}$). A non-linear transformation using the spatial-normalization

algorithm (implemented in SPM8; *Friston et al., 1994*) was employed to transform individual MRIs to the standard MNI brain. The source maps were plotted using the Surf Ice tool (https://www.nitrc.org/projects/surface/).

Furthermore, we selected predefined ROIs for the subsequent analysis using the Brainnetome atlas (*Fan et al., 2016*). The ROIs included three areas within the left STG (BA 41/42 of the auditory cortex, rostral portion of BA 22, and caudal portion of BA 22) and left IFG (dorsal, ventral, and opercular portions of BA 44). We created an LCMV filter using the same parameters used for whole-brain source reconstruction. The virtual electrode in the source space corresponding to each ROI was generated. Singular vector decomposition was applied to select the component with maximum variance. Later, the differential omission MMN was computed by subtracting the power of the long tones from the power of the long omissions, and the power of the short tones from the power of the short omissions. This procedure was applied to each group separately (i.e., Spanish and Basque natives).

To examine the presence of an interaction effect at the source level, we first averaged the source activity time-series across ROIs for both the STG and IFG. A cluster-based permutation test was then applied on source activity data to identify temporal clusters over the 0.100–0.250 s time interval, following the same contrasts that were performed at the sensor level. This was done separately for the STG and IFG. This approach is similar to the one performed at the sensor level, but provides higher temporal sensitivity by considering individual time points within the 0.100–0.250 s interval. All the remaining parameters of the cluster-based permutation test are the same as the test used for the sensor-level analysis. When a cluster associated with a significant p-value was detected, the highest peak within the cluster was selected and used for the subsequent pairwise comparisons. Between-group comparisons were performed on each ROI and peak using a one-sided independent sample *t*-test, following the same contrasts employed at the sensor level.

## Acknowledgements

This research was supported by the Basque Government through the BERC 2022–2025 program and by the Spanish State Research Agency through BCBL Severo Ochoa excellence accreditation CEX2020-001010-S. Work by PM received support from 'la Caixa' Foundation (ID 100010434) through the fellowship LCF/BQ/IN17/11620019, and the European Union's Horizon 2020 research and innovation programme under the Marie Skłodowska-Curie grant agreement no. 713673. CM received funding from the European Research Council (ERC) under the European Union's Horizon 2020 research and innovation programme (Grant Agreement No: 819093), the Spanish Ministry of Economy and Competitiveness (PID2020-113926GB-I00), and the Basque Government (PIBA18_29). NM was supported by the Spanish Ministry of Economy and Competitiveness (PSI2015-65694-P, RTI2018-096311-B-I00, PDC2022-133917-I00). Work by ML received support from Juan de la Cierva IJC2020-042886-I. SN acknowledges the support from 'The Adaptive Mind', funded by the Excellence Program of the Hessian Ministry of Higher Education, Science, Research and Art. We wish to express our gratitude to the BCBL lab staff and the research assistants who helped to recruit the participants and collect the data. We thank Ram Frost for providing helpful comments on the manuscript.

## Additional information

### Funding

| Funder | Grant reference number | Author |
| --- | --- | --- |
| 'la Caixa' Foundation | LCF/BQ/IN17/11620019 | Piermatteo Morucci |
| H2020 Marie Skłodowska-Curie Actions | 10.3030/713673 | Piermatteo Morucci |
| Ministerio de Asuntos Económicos y Transformación Digital, Gobierno de España | PSI2015-65694-P | Nicola Molinaro |

| Funder | Grant reference number | Author |
|---|---|---|
| Ministerio de Asuntos Económicos y Transformación Digital, Gobierno de España | RTI2018-096311-B-I00 | Nicola Molinaro |
| Ministerio de Asuntos Económicos y Transformación Digital, Gobierno de España | PDC2022-133917-I00 | Nicola Molinaro |
| Agencia Estatal de Investigación | CEX2020-001010-S | Nicola Molinaro |
| European Research Council | 819093 | Clara Martin |
| Ministerio de Ciencia e Innovación | IJC2020-042886-I | Mikel Lizarazu |
| Spanish Ministry of Economy and Competitiveness | PID2020-113926GB-I00 | Clara Martin |
| Basque Government | PIBA18_29 | Clara Martin |

The funders had no role in study design, data collection, and interpretation, or the decision to submit the work for publication.

### Author contributions

Piermatteo Morucci, Conceptualization, Data curation, Software, Formal analysis, Validation, Investigation, Visualization, Methodology, Writing - original draft, Project administration, Writing - review and editing; Sanjeev Nara, Formal analysis, Methodology, Writing - review and editing; Mikel Lizarazu, Supervision, Visualization, Methodology; Clara Martin, Conceptualization, Supervision, Writing - review and editing; Nicola Molinaro, Conceptualization, Resources, Formal analysis, Supervision, Funding acquisition, Visualization, Methodology, Project administration, Writing - review and editing

### Author ORCIDs

Piermatteo Morucci ⓘ https://orcid.org/0000-0002-4972-0864
Nicola Molinaro ⓘ https://orcid.org/0000-0002-7549-6042

### Ethics

Participants were recruited through the participant recruitment system of the Basque Center on Cognition, Brain and Language. The experiment and methods received approval from both the ethical committee and scientific committee of the Basque Center on Cognition, Brain and Language (Ethics approval number: 18072018M), following with the principles of the Declaration of Helsinki. Written informed consent was obtained from all participants in line with the guidelines of the Research Committees of the BCBL.

Reviewer #1 (Public Review): https://doi.org/10.7554/eLife.91636.3.sa1
Reviewer #2 (Public Review): https://doi.org/10.7554/eLife.91636.3.sa2
Reviewer #3 (Public Review): https://doi.org/10.7554/eLife.91636.3.sa3
Author response https://doi.org/10.7554/eLife.91636.3.sa4

## Additional files

### Supplementary files
• MDAR checklist

### Data availability

Data and codes for reproducing the analysis and figures are publicly available via the Open Science Framework (OSF): https://osf.io/6jep8/.

The following dataset was generated:

| Author(s) | Year | Dataset title | Dataset URL | Database and Identifier |
|---|---|---|---|---|
| Morucci P, Molinaro N | 2023 | Language experience shapes predictive coding of rhythmic sound sequences | https://osf.io/6jep8/ | Open Science Framework, osf.io/6jep8 |

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
