## [Editor Report · eLife assessment]

This study presents **important** observations about how the human brain uses long-term priors (acquired during our lifetime of listening) to make predictions about expected sounds - an open question in the field of predictive processing. The evidence presented is **solid** and based on state-of-the-art statistical analysis, but limited by a relatively low N and low magnitude for the interaction effect.

---

## [Referee Report · Reviewer #1 (Public Review)]

Summary:

In this work, the authors study whether the human brain uses long term priors (acquired during our lifetime) regarding the statistics of auditory stimuli to make predictions respecting auditory stimuli. This is an important open question in the field of predictive processing.

To address this question, the authors cleverly profit from the naturally existing differences in two linguistic groups. While speakers of Spanish use phrases in which function-words (short words like, articles and prepositions) are followed by content-words (longer words like nouns, adjectives and verbs), speakers of Basque use phrases with the opposite order. Because of this, speakers of Spanish usually hear phrases in which short words are followed by longer words, and speakers of Basque experience the opposite. This difference in the order of short and longer words is hypothesized to result in a long term duration prior that is used to make predictions regarding the likely durations of incoming sounds, even if they are not linguistic in nature.

To test this, the authors used MEG to measure the mismatch responses (MMN) elicited by the omission of short and long tones that were presented in alternation. The authors report an interaction between the language background of the participants (Spanish, Basque) and the type of omission MMN (short, long), which goes in line with their predictions. They supplement these results with a source level analysis.

Strengths:

This work has many strengths. To test the main question, the authors profit from naturally occurring differences in the everyday auditory experiences of two linguistic groups, which allows to test the effect of putative auditory priors consolidated over the years. This is a direct way of testing the effect of long term priors.

The fact that the priors in question are linguistic and that the experiment was conducted using non-linguistic stimuli (i.e. simple tones), allows to test if these long term priors generalize across auditory domains.

The experimental design is elegant and the analysis pipeline appropriate. This work is very well written. In particular the introduction and discussion sections are clear and engaging. The literature review is complete.

Weaknesses:

The authors report a widespread omission response, which resembles the classical mismatch response (in MEG planar gradiometers) with strong activations in sensors over temporal regions. However the interaction reported is circumscribed to four sensors that do not overlap with the peaks of activation of the omission response.

---

## [Referee Report · Reviewer #2 (Public Review)]

Summary:

Morucci et al. tested the influence of linguistic prosody long-term priors in forming predictions about simple acoustic rhythmic tone sequences composed of alternating tone duration, by violating context-dependent short-term priors formed during sequence listening. Spanish and Basque participants were selected due to the different rhythmic prosody of the two languages (functor-initial vs. Functor final, respectively), despite a common cultural background. The authors found that neuromagnetic responses to casual tone omissions reflected the linguistic prosody pattern of the participant's dominant language: in Spanish speakers, omission responses were larger to short tones, whereas in Basque speakers, omission responses were larger to long tones. Source localization of these responses revealed this interaction pattern in the left auditory cortex, which the authors interpret as reflecting a perceptual bias due to acoustic cues (inherent linguistic rhythms, rather than linguistic content). Importantly, this pattern was not found when the rhythmic sequence entailed pitch, rather than duration, cues. To my knowledge, this is the first study providing neural signatures of a known behavioral effect linking ambiguous rhythmic tone sequence perceptual organization to linguistic experience.

The conclusions of the study are well supported by the data. The hypotheses, albeit allowing alternative perspectives, are well justified according to the existing literature. Albeit with inconclusive results, additional analyses to test entrained oscillatory activity to the perceived rhythms have been performed, which adds explanatory power to the study.

Strengths:

(1) The choice of participants. The bilingual population of the Basque country is perfect for performing studies which need to control for cultural and socio-economic background while having profound linguistic differences. In this sense, having dominant Basque speakers as a sample equates that in Molnar et al. (2016), and thus overcomes the lack of direct behavioral evidence for a difference in rhythmic grouping across linguistic groups. Molnar et al. (2016)'s evidence on the behavioral effect is compelling, and the evidence on neural signatures provided by the present study aligns with it.

(2) The experimental paradigm. It is a well designed acoustic sequence, which considers aspects such as gap length insertion, to be able to analyze omission responses free from subsequent stimulus-driven responses, and which includes a control sequence which uses pitch instead of duration as a cue to rhythmic grouping, which provides a stronger case for the differences found between groups to be due to prosodic duration cues.

(3) Data analyses. Sound, state-of-the-art methodology in the event-related field analyses at the sensor and source levels.

Weaknesses:

(1) The main conclusion of the study reflects a known behavioral effect on rhythmic sequence perceptual organization driven by linguistic background (Molnar et al. 2016, particularly) and, thus, the novelty of the findings is restricted to neural activity evidence.

(2) Although the paradigm is well designed, there are alternative views in formulating the hypotheses. For instance, one could argue that, according to predictive coding views, omission responses should be larger when the gap occurs at the end of the pattern, as that would be where stronger expectations are placed. However, the authors provide good justification based on previous literature for the expectation of larger omission responses at the downbeat of a rhythmic pattern.

---

## [Referee Report · Reviewer #3 (Public Review)]

Summary:

The paper investigates the effects of long-term linguistic experience on early auditory processing, a subject that has been relatively less studied compared to short-term influences. Using MEG, the study examines brain responses to auditory stimuli in speakers of Spanish and Basque, whose syntactic rules provide different degrees of exposure to durational patterns (long-short vs short-long). The findings suggest that both long-term language experience as well as short-term transitional probabilities can shape auditory predictive coding for non-linguistic sound sequences, evidenced by differences in mismatch negativity amplitudes localised to left auditory cortex.

Strengths:

The study integrates linguistics and auditory neuroscience in an interesting interdisciplinary way that may interest linguists as well as neuroscientists. The fact that long-term language experience affects early auditory predictive coding is important for understanding group and individual differences in domain-general auditory perception. It has importance for neurocognitive models of auditory perception (e.g. inclusion of long-term priors), and will be of interest to researchers in linguistics, auditory neuroscience, and the relationship between language and perception. The inclusion of a control condition based on pitch is also a strength.

Weaknesses:

The main weaknesses are the strength of the effects and generalisability. Only two languages were examined, Spanish and Basque. The sample size is also relatively small by today's standards, with N=20 in each group. Furthermore, the crucial effects are all mostly in the .01>P<.05 range, such as the crucial interaction P=.03, although I note the methods used to derive the results are sound and state-of-the-art. It would be nice to see it replicated in the future, with more participants and other languages. It would also have been nice to see behavioural data that could be correlated with neural data to better understand the real-world consequences of the effect.

---

## [Author Response]

The following is the authors’ response to the current reviews.

We thank the Reviewers and Editors for the constructive comments, which we believe have significantly improved the quality of our manuscript.

The following is the authors’ response to the original reviews.

**Reviewer #1 (Public Review):**
(1) With respect to the predictions, the authors propose that the subjects, depending on their linguistic background and the length of the tone in a trial, can put forward one or two predictions. The first is a short-term prediction based on the statistics of the previous stimuli and identical for both groups (i.e. short tones are expected after long tones and vice versa). The second is a long-term prediction based on their linguistic background. According to the authors, after a short tone, Basque speakers will predict the beginning of a new phrasal chunk, and Spanish speakers will predict it after a long tone.In this way, when a short tone is omitted, Basque speakers would experience the violation of only one prediction (i.e. the short-term prediction), but Spanish speakers will experience the violation of two predictions (i.e. the short-term and long-term predictions), resulting in a higher amplitude MMN. The opposite would occur when a long tone is omitted. So, to recap, the authors propose that subjects will predict the alternation of tone durations (short-term predictions) and the beginning of new phrasal chunks (long-term predictions).The problem with this is that subjects are also likely to predict the completion of the current phrasal chunk. In speech, phrases are seldom left incomplete. In Spanish is very unlikely to hear a function-word that is not followed by a content-word (and the opposite happens in Basque). On the contrary, after the completion of a phrasal chunk, a speaker might stop talking and a silence might follow, instead of the beginning of a new phrasal chunk.Considering that the completion of a phrasal chunk is more likely than the beginning of a new one, the prior endowed to the participants by their linguistic background should make us expect a pattern of results actually opposite to the one reported here.

We thank the Reviewer #1 for this pertinent comment and the opportunity to address this issue. A very similar concern was also raised by Reviewer #2. Below we try to clarify the motivations that led us to predict that the hypothesized long-term predictions should manifest at the onset (and not within or the end) of a perceptual chunk.

Reviewers #1 and #2 contest a critical assumption of our study i.e., the fact that longterm predictions should occur at the beginning of a rhythmic chunk as opposed to its completion. They also contest the prediction deriving from this view i.e., omitting the first sound in a perceptual chunk (short for Spanish, long for Basque) would lead to larger error responses than omitting a later element. They suggest an alternative view: the omission of tones at the end of a perceptual rhythmic chunk would evoke larger error responses than omissions at its onset, as subjects are more likely to predict the completion of the chunk than its beginning. This view predicts an interaction effect in the opposite direction of our findings.

While we acknowledge this as a plausible hypothesis, we believe that the current literature provides strong support for our view. Indeed, many studies in the rhythm and music perception literature have investigated the ERP responses to deviant sounds and omissions placed at different positions within rhythmic patterns (e.g., Ladinig et al., 2009; Bouwer et al., 2016; Brochard et al., 2003; Potter et al., 2009; Yabe et al., 2001). For instance, Lading et al., 2009 presented participants with metrical rhythmical sound sequences composed of eight tones. In some deviant sequences, the first or a later tone was omitted. They found that earlier omissions elicited earlier and higher-amplitude MMN responses than later omissions (irrespective of attention). Overall, this and other studies showed that the amplitude of ERP responses are larger when deviants occur at positions that are expected to be the “start” of a perceptual group - “on the beat” in musical terms - and decline toward the end of the chunk. According to some of these studies, the first element of a chunk is particularly important to track the boundaries of temporal sequences, which is why more predictive resources are invested at that position. We believe that this body of evidence provides robust bases for our hypotheses and the directionality of our predictions.

An additional point that should be considered concerns the amplitude of the prediction error response elicited by the omission. From a predictive coding perspective, the omission of the onset of a chunk should elicit larger error responses because the system is expecting the whole chunk (i.e., two tones/more acoustic information). On the other hand, the omission of the second tone - in the transition between two tones within the chunk - should elicit a smaller error response because the system is expecting only the missing tone (i.e. less acoustic information).

Given the importance of these points, we have now included them in the updated version of the paper, in which we try to better clarify the rationale behind our hypothesis (see Introduction section, around the 10th paragraph).

(2) The authors report an interaction effect that modulates the amplitude of the omission response, but caveats make the interpretation of this effect somewhat uncertain. The authors report a widespread omission response, which resembles the classical mismatch response (in MEG) with strong activations in sensors over temporal regions. Instead, the interaction found is circumscribed to four sensors that do not overlap with the peaks of activation of the omission response.

We thank the Reviewer for this comment. As mentioned in the provisional response, the approach employed to identify the presence of an interaction effect was conservative: We utilized a non-parametric test on combined gradiometers data, without making a priori assumptions about the location of the effect, and employed small cluster thresholds (cfg.clusteralpha = 0.05) to increase the chances of detecting highly localized clusters with large effect sizes. The fact that the interaction effect arises in a relatively small cluster of sensors does not alter its statistical robustness. It should be also considered that in the present analyses we focused on planar gradiometer data that, compared to magnetometers and axial gradiometers, present more fine-grained spatial resolution and are more suited for picking up relatively small effects.

The partial overlap of the cluster with the activation peaks may simply reflect the fact that different sources contribute to the generation of the omission-MMN, which has been reported in several studies (e.g., Zhang et al., 2018; Ross & Hamm, 2020). We value the Reviewer’s input and are grateful for the opportunity to address these considerations.

Furthermore, the boxplot in Figure 2E suggests that part of the interaction effect might be due to the presence of two outliers (if removed, the effect is no longer significant). Overall, it is possible that the reported interaction is driven by a main effect of omission type which the authors report, and find consistently only in the Basque group (showing a higher amplitude omission response for long tones than for short tones). Because of these points, it is difficult to interpret this interaction as a modulation of the omission response.

We thank the Reviewer for the comment and appreciate the opportunity to address these concerns. We have re-evaluated the boxplot in Figure 2E and want to clarify that the two participants mentioned by Reviewer #1, despite being somewhat distant from the rest of the group, are not outliers according to the standard Tukey’s rule. As shown in the figure below, no participant fell outside the upper (Q3+1.5xIQR) and lower whiskers (Q1-1.5xIQR) of the boxplot.

Moreover, we believe that the presence of a main effect of omission type does not impact the interpretation of the interaction, especially considering that these effects emerge over distinct clusters of channels (see Fig. 1 C; Supplementary Fig. 2 A).

Based on these considerations - and along with the evidence collected in the control study and the source reconstruction data reported in the new version of the manuscript - we find it unlikely that the interaction effect is driven by outliers or by a main effect of omission type. We appreciate the opportunity provided by the Reviewer to address these concerns, as we believe they strengthen the claim that the observed effect is driven by the hypothesized long-term linguistic priors rather than uncontrolled group differences.

**Author response image 1. sa4fig1:** 

It should also be noted that in the source analysis, the interaction only showed a trend in the left auditory cortex, but in its current version the manuscript does not report the statistics of such a trend.

We appreciate the Reviewer’s suggestion to incorporate more comprehensive source analyses. In the new version of the paper, we perform new analyses on the source data using a new Atlas with more fine-grained parcellations of the regions of interests (ROIs) (Brainnetome atlas; Fan et al., 2016) and focusing on peak activity to increase response’s sensitivity in space and time. We therefore invite the Reviewer to read the updated part on source reconstruction included in the Results and Methods sections of the paper.

**Reviewer #1 (Recommendations For The Authors):**
While I have described my biggest concerns with respect to this work in the public review, here I list more specific points that I hope will help to improve the manuscript. Some of these are very minor, but I hope you will still find them constructive.(1) I understand the difficulties implied in recruiting subjects from two different linguistic groups, but with 20 subjects per group and a between-groups design, the current study is somewhat underpowered. A post-hoc power analysis shows an achieved power of 46% for medium effect sizes (d = 0.5, and alpha = 0.05, one-sided test). A sensitivity analysis shows that the experiment only has 80% power for effect sizes of d = 0.8 and above. It would be important to acknowledge this limitation in the manuscript.

We thank the Reviewer for reporting these analyses. It must be noted that our effect of interest was based on Molnar et al.’s (2016) behavioral experiment, in which a sample size of 16 subjects per group was sufficient to detect the perceptual grouping effect. In Yoshida et al., (2010), the perceptual grouping effect emerged with two groups of 20 7–8-month-old Japanese and English-learning infants. Based on these previous findings, we believe that a sample size of 20 participants per group can be considered appropriate for the current MEG study. We clarified these aspects in the Participants section of the manuscript, in which we specified that previous behavioral studies detected the perceptual grouping with similar sample sizes. Moreover, to acknowledge the limitation highlighted by the Reviewer, we also include the power and sensitivity analysis in a note in the same section (see note 2 in the Participants section).

(2) All the line plots in the manuscript could be made much more informative by adding 95% CI bars. For example, in Figure 4A, the omission response for the long tone departs from the one for the short tone very early. Adding CIs would help to assess the magnitude of that early difference. Error bars are present in Figure 3, but it is not specified what these bars represent.

Thanks for the comments. We added the explanation of the error bars in the new version of Figure 3. For the remaining figures, we prefer maintaining the current version of the ERF, as the box-plots accompanying them provide information about the distribution of the effect across participants.

(3) In the source analysis, there is only mention of an interaction trend in the left auditory cortex, but no statistics are presented. If the authors prefer to mention such a trend, I think it would be important to provide its stats to allow the reader to assess its relevance.

We performed new analysis on the source data, all reported in the updated version of the manuscript.

(4) In the discussion section, the authors refer to the source analysis and state that "the interaction is evident in the left". But if only a statistical trend was observed, this statement would be misleading.

We agree with this comment. We invite the Reviewer to check the new part on source reconstruction, in which contrasts going in the same direction of the sensor level data are performed.

(5) In the discussion the authors argue that "This result highlights the presence of two distinct systems for the generation of auditory" that operate at different temporal scales, but the current work doesn't offer evidence for the existence of two different systems. The effects of long-term priors and short-term priors presented here are not dissociated and instead sum up. It remains possible that a single system is in place, collecting statistics of stimuli over a lifetime, including the statistics experienced during the experiment.

Thanks for pointing that out. We changed the sentence above as follows: “This result highlights the presence of an active predictive system that relies on natural sound statistics learned over a lifetime to process incoming auditory input”.

(6) In the discussion, the authors acknowledge that the omission response has been interpreted both as pure prediction and as pure prediction error. Then they declare that "Overall, these findings are consistent with the idea that omission responses reflect, at least in part, prediction error signals.". However an argument for this statement is not provided.

Thanks for pointing out this lack of argument. In the new version of the manuscript, we explained our rationale as follows: “Since sensory predictive signals primarily arise in the same regions as the actual input, the activation of a broader network of regions in omission responses compared to tones suggests that omission responses reflect, at least in part, prediction error signals”.

(7) In the discussion the authors present an alternative explanation in which both groups might devote more resources to the processing of long events, because these are relevant content words. Following this, they argue that "Independently on the interpretation, the lack of a main effect of omission type in the control condition suggests that the long omission effect is driven by experience with the native language." However as there was no manipulation of duration in the control experiment, a lack of the main effect of omission type there does not rule out the alternative explanation that the authors put forward.

This is correct; thanks for noticing it. We removed the sentence above to avoid ambiguities.

Minor points:(8) The scale of the y-axis in Figure 2C might be wrong, as it goes from 9 to 11 and then to 12. If the scale is linear, the top value should be 13, or the bottom value should be 10.

Figure 2C has been modified accordingly, thanks for noticing the error.

(9) There is a very long paragraph starting on page 7 and ending on page 8. Toward the end of the paragraph, the analysis of the control condition is presented. That could start a new paragraph.

Thanks for the suggestion. We modified the manuscript as suggested.

**Reviewer #2 (Public Review):**
(1) Despite the evidence provided on neural responses, the main conclusion of the study reflects a known behavioral effect on rhythmic sequence perceptual organization driven by linguistic background (Molnar et al. 2016, particularly). Also, the authors themselves provide a good review of the literature that evidences the influence of longterm priors in neural responses related to predictive activity. Thus, in my opinion, the strength of the statements the authors make on the novelty of the findings may be a bit far-fetched in some instances.

Thanks for the suggestion. A similar point was also advanced by Reviewer 1. In general, we believe our work speaks about the predictive nature of such experiencedependent effects, and show that these linguistic priors shape sensory processes at very early stages. This is discussed in the sixth and seventh paragraphs of the Discussion section. In the new version of the article, we modified some statements and tried to make them more coherent with the scope of the present work. For instance, we changed "This result highlights the presence of two distinct systems for the generation of auditory predictive models, one relying on the transition probabilities governing the recent past, and another relying on natural sound statistics learned over a lifetime“ with “This result highlights the presence of an active predictive system that relies on natural sound statistics learned over a lifetime to process incoming auditory input”.

(2) Albeit the paradigm is well designed, I fail to see the grounding of the hypotheses laid by the authors as framed under the predictive coding perspective. The study assumes that responses to an omission at the beginning of a perceptual rhythmic pattern will be stronger than at the end. I feel this is unjustified. If anything, omission responses should be larger when the gap occurs at the end of the pattern, as that would be where stronger expectations are placed: if in my language a short sound occurs after a long one, and I perceptually group tone sequences of alternating tone duration accordingly, when I hear a short sound I will expect a long one following; but after a long one, I don't necessarily need to expect a short one, as something else might occur.

A similar point was advanced by Reviewer #1. We tried to clarify the rationale behind our hypothesis. Please refer to the response provided to the first comment of Reviewer #1 above.

(3) In this regard, it is my opinion that what is reflected in the data may be better accounted for (or at least, additionally) by a different neural response to an omission depending on the phase of an underlying attentional rhythm (in terms of Large and Jones rhythmic attention theory, for instance) and putative underlying entrained oscillatory neural activity (in terms of Lakatos' studies, for instance). Certainly, the fact that the aligned phase may differ depending on linguistic background is very interesting and would reflect the known behavioral effect.

We thank the Reviewer for this comment. We explored in more detail the possibility that the aligned phase may differ depending on linguistic background, which is indeed a very interesting hypothesis. In the phase analyses reported below we focused on the instantaneous phase angle time locked to the onset of short and long tones presented in the experiment.

In short, we extracted time intervals of two seconds centered on the onset of the tones for each participant (~200 trials per condition) and using a wavelet transform (implemented in Fieldtrip ft_freqanalysis) we targeted the 0.92 Hz frequency that corresponds to the rhythm of presentation of our pairs of tones. We extracted the phase angle for each time point and using the circular statistics toolbox implemented in Matlab we computed the Raleigh z scores across all the sensor space for each tone (long and short tone) and group (Spanish (Spa) dominants and Basque (Eus) dominants). This method evaluates the instantaneous phase clustering at a specific time point, thus evaluating the presence of a specific oscillatory pattern at the onset of the specific tone.

**Author response image 2. sa4fig2:** 

Here we observe that the phase clustering was stronger in the right sensors for both groups. The critical point is to evaluate the phase angle (estimated in phase radians) for the two groups and the two tones and see if there are statistical differences. We focused first on the sensor with higher clustering (right temporal MEG1323) and observed very similar phase angles for the two groups both for long and short tones (see image below). We then focused on the four left fronto-temporal sensor pairs who showed the significant interaction: here we observed one sensor (MEG0412) with different effects for the two groups (interaction group by tone was significant, p=0.02): for short tones the “Watson (1961) approximation U2 test” showed a p-value of 0.11, while for long tones the p-value was 0.03 (after correction for multiple comparisons).

Overall, the present findings suggest the tendency to phase aligning differently in the two groups to long and short tones in the left fronto-temporal hemisphere. However, the effect could be detected only in one gradiometer sensor and it was not statistically robust. The effect in the right hemisphere was statistically more robust, but it was not sensitive to group language dominance.

Due to the inconclusive nature of these analyses regarding the role of language experience in shaping the phase alignment to rhythmic sound sequences, we prefer to keep these results in the public review rather than incorporating them in the article. Nonetheless, we believe that this decision does not undermine the main finding that the group differences in the MMN amplitude are driven by long-term predictions – especially in light of the many studies indicating the MMN as a putative index of prediction error (e.g., Bendixen et al., 2012; Heilbron and Chait, 2018). Moreover, as suggested in the preliminary reply, despite evoked responses and oscillations are often considered distinct electrophysiological phenomena, current evidence suggests that these phenomena are interconnected (e.g., Studenova et al., 2023). In our view, the hypotheses that the MMN reflects differences in phase alignment and long-term prediction errors are not mutually exclusive.

**Author response image 3. sa4fig3:** 

(4) Source localization is performed on sensor-level significant data. The lack of sourcelevel statistics weakens the conclusions that can be extracted. Furthermore, only the source reflecting the interaction pattern is taken into account in detail as supporting their hypotheses, overlooking other sources. Also, the right IFG source activity is not depicted, but looking at whole brain maps seems even stronger than the left. To sum up, source localization data, as informative as it could be, does not strongly support the author's claims in its current state.

A similar comment was also advanced by Reviewer #1 (comment 2). We appreciate the suggestion to incorporate more comprehensive source analyses. In the new version of the paper, we perform new analyses on the source data using a new Atlas with more fine-grained parcellations of the ROIs, and focusing on peak activity to increase response’s sensitivity in space and time. We therefore invite the Reviewer to read the updated part on source reconstruction included in the Results and Methods sections of the paper.

In the article, we report only the source reconstruction data from ROIs in the left hemisphere, because it is there that the interaction effect arises at the sensor level. However, we also explored the homologous regions in the right hemisphere, as requested by the Reviewer. A cluster-based permutation test focusing on the interaction between language group and omission type was performed on both the right STG and IFG data. No significant interaction emerged in any of these regions. Below a plot of the source activity time series over ROIs in the right STG and IFG.

**Author response image 4. sa4fig4:** 

**Reviewer #2 (Recommendations For The Authors):**
In this set of private recommendations for the authors, I will outline a couple of minor comments and try to encourage additional data analyses that, in my opinion, would strengthen the evidence provided by the study.(1) As I noted in the public review, I believe an oscillatory analysis of the data would, on one hand, provide stronger support for the behavioral effect of rhythmic perceptual organization given the lack of behavioral direct evidence; and, on the other hand, provide evidence (to be discussed if so) for a role of entrained oscillation phase in explaining the different pattern of omission responses. One analysis the authors could try is to measure the phase angle of an oscillation, the frequency of which relates to the length of the binary pattern, at the onset of short and long tones, separately, and compare it across groups. Also, single trials of omission responses could be sorted according to that phase.

Thanks for the suggestion. Please see phase analyses reported above.

(2) I wonder why source activity for the right IFG was not shown. I urge the authors to provide and discuss a more complete picture of the source activity found. Given the lack of source statistics (which could be performed), I find it a must to give an overall view. I find it so because I believe the distinction between perceptual grouping effects due to inherent acoustic differences across languages or semantic differences is so interesting.

Thanks again for the invitation to provide a more complete picture of the source activity data. As mentioned in the response above, we invite the Reviewer to read the new related part included in the Results and Methods sections of the paper. In our updated source reconstruction analysis, we find that some regions around the left STG show a pattern that resembles the one found at the sensor-level, providing further support for the “acoustic” (rather than syntactic/semantic) nature of the effect.

We did not report ROI analysis on the right hemisphere because the interaction effect at sensor level emerged on the left hemisphere. Yet, we included a summary of this analysis in the public response above.

(3) Related to this, I have to acknowledge I had to read the whole Molnar et al. (2016) study to find the only evidence so far that, acoustically, in terms of sound duration, Basque and Spanish differ. This was hypothesized before but only at Molnar, an acoustic analysis is performed. I think this is key, and the authors should give it a deeper account in their manuscript. I spend my review of this study thinking, well, but when we speak we actually bind together different words and the syllabic structure does not need to reflect the written one, so maybe the effect is due to a high-level statistical prior related to the content of the words... but Molnar showed me that actually, acoustically, there's a difference in accent and duration: "Taken together, Experiments 1a and 1b show that Basque and Spanish exhibit the predicted differences in terms of the position of prosodic prominence in their phonological phrases (Basque: trochaic, Spanish: iambic), even though the acoustic realization of this prominence involves not only intensity in Basque but duration, as well. Spanish, as predicted, only uses duration as a cue to mark phrasal prosody."

Thanks for the suggestion, the distinction in terms of sound duration in Spanish and Basque reported by Molnar is indeed very relevant for the current study.

We add a few sentences to highlight the acoustic analysis by Molnar and the consequent acoustic nature of the reported effect.

In the introduction: “Specifically, the effect has been proposed to depend on the quasiperiodic alternation of short and long auditory events in the speech signal – reported in previous acoustic analyses (Molnar et al., 2016) – which reflect the linearization of function words (e.g., articles, prepositions) and content words (e.g., nouns, adjectives, verbs).”

In the discussion, paragraph 3, we changed “We hypothesized that this effect is linked to a long-term “duration prior” originating from the syntactic function-content word order of language, and specifically, from its acoustic consequences on the prosodic structure” with “We hypothesized that this effect is linked to a long-term “duration prior” originating from the acoustic properties of the two languages, specifically from the alternation of short and long auditory events in their prosody”.

In the discussion, end of paragraph eight: “The reconstruction of cortical sources associated with the omission of short and long tones in the two groups showed that an interaction effect mirroring the one at the sensor level was present in the left STG, but not in the left IFG (fig. 3, B, C, D). Pairwise comparisons within different ROIs of the left STG indicated that the interaction effect was stronger over primary (BA 41/42) rather than associative (BAs 22) portions of the auditory cortex. Overall, these results suggest that the “duration prior” is linked to the acoustic properties of a given language rather than its syntactic configurations”.

Now, some minor comments:(1) Where did the experiments take place? Were they in accordance with the Declaration of Helsinki? Did participants give informed consent?

All the requested information has been added to the updated version of the manuscript. Thanks for pointing out this.

(2) The fixed interval should be called inter-stimulus interval.

Thanks for pointing this out. We changed the wording as suggested.

(3) The authors state that "Omission responses allow to examine the presence of putative error signals decoupled from bottom-up sensory input, offering a critical test for predictive coding (Walsh et al 2020, Heilbron and Chait, 2018).". However the way omission responses are computed in their study is by subtracting the activity from the previous tone. This necessarily means that in the omission activity analyzed, there's bottom-up sensory input activity. As performing another experiment with a control condition in which a sequence of randomly presented tones with different durations to compare directly the omission activity in both sequences (experimental and control) is possibly too demanding, I at least urge the authors to incorporate the fact that their omission responses do reflect also tone activity. And consider, for future experiments, the inclusion of further control conditions.

Thanks for the opportunity to clarify this aspect. Actually, the way we computed the omission MMN is not by subtracting the activity of the previous tone from the omission, but by subtracting the activity of randomly selected tones across the whole experiment. That is, we randomly selected around 120 long and short tones (i.e., about the same number as the omissions); we computed the ERF for the long and short tones; we subtracted these ERF from the ERF of the corresponding short and long omissions. We clarified these aspects in both the Materials and Methods (ERF analysis paragraph) and Results section.

Moreover, the subtraction strategy - which is the standard approach to calculate the MMN - allows to handle possible neural carryover effects arising from the perception of the tone preceding the omission.

The sentence "Omission responses allow to examine the presence of putative error signals decoupled from bottom-up sensory input, offering a critical test for predictive coding (Walsh et al 2020, Heilbron and Chait, 2018)." simply refer to the fact that the error responses resulting from an omission are purely endogenous, as omissions are just absence of an expected input (i.e., silence). On the other hand, when a predicted sequence of tones is disrupted by an auditory deviants (e.g., a tone with a different pitch or duration than the expected one), the resulting error response is not purely endogenous, but it partially includes the response to the acoustic properties of the deviant.

(4) When multiple clusters emerged from a comparison, only the most significant cluster was reported. Why?

We found more than one significant cluster only in the comparison between pure omissions vs tones (figure 2 A, B). The additional significant cluster from this comparison is associated with a P-value of 0.04, emerges slightly earlier in time, and goes in the same direction as the cluster reported in the paper i.e., larger ERF responses for omission vs tones. We added a note specifying the presence of this second cluster, along with a figure on the supplementary material (Supplementary Fig. 1 A, B).

(5) Fig 2, if ERFs are baseline corrected -50 to 0ms, why do the plots show pre-stimulus amplitudes not centered at 0?

This is because we combined the latitudinal and longitudinal gradiometers on the ERF obtained after baseline correction, by computing the root mean square of the signals at each sensor position (see also https://www.fieldtriptoolbox.org/example/combineplanar_pipelineorder/). This information is reported in the methods part of the article.

(6) Fig 2, add units to color bars.

Sure.

(7) Fig 2 F and G, put colorbar scale the same for all topographies.

Sure, thanks for pointing this out.

(8) The interaction effect language (Spanish; Basque) X omission type (short; long) appears only in a small cluster of 4 sensors not located at the locations with larger amplitudes to omissions. Authors report it as left frontotemporal, but it seems to me frontocentral with a slight left lateralization.

(1) the fact that the cluster reflecting the interaction effect does not overlap with the peaks of activity is not surprising in our view. Many sources contribute to the generation of the MMN. The goal of our work was to establish whether there is also evidence for a long-term system (among the many) contributing to this. That is why we perform a first analysis on the whole omission response network (likely including many sources and predictive/attentional systems), and then we zoom in and focus on our hypothesized interaction. We never claim that the main source underlying the omissionMMM is the long-term predictive system.

(2) The exact location of those sensors is at the periphery of the left-hemisphere omission response, which mainly reflects activity from the left temporal regions. The sensor location of this cluster could be influenced by multiple factors, including (i) the direction of the source dipoles determining an effect; (ii) the combination of multiple sources contributing to the activity measured at a specific sensor location, whose unmixing could be solved only with a beamforming source approach. Based on the whole evidence we collected also in the source analyzes we concluded that the major contributors to the sensor-level interaction are emerging from both frontal and temporal regions.

**Reviewer #3 (Public Review):**

(1) The main weaknesses are the strength of the effects and generalisability. The sample size is also relatively small by today's standards, with N=20 in each group. Furthermore, the crucial effects are all mostly in the .01>P<.05 range, such as the crucial interaction P=.03. It would be nice to see it replicated in the future, with more participants and other languages. It would also have been nice to see behavioural data that could be correlated with neural data to better understand the real-world consequences of the effect.

We appreciate the positive feedback from Reviewer #3. We agree that it would be nice to see this study replicated in the future with larger sample sizes and a behavioral counterpart. Below are a few comments concerning the weakness highlighted:

(i) Concerning the sample size: a similar point was raised by Reviewer #1. We report our reply as presented above: “Despite a sample size of 20 participants per group can be considered relatively small for detecting an effect in a between-group design, it must be noted that our effect of interest was based on Molnar et al.’s (2016) experiment, where a sample size of 16 subjects per group was sufficient to detect the perceptual grouping effect. In Yoshida et al., 2010, the perceptual grouping effect arose with two groups of 20 7–8-month-old Japanese and English-learning infants. Based on these findings, we believe that a sample size of 20 participants per group can be considered appropriate for the current study”. We clarified these aspects in the new version of the manuscript.

(ii) We believe that the lack of behavioral data does not undermine the main findings of this study, given the careful selection of the participants and the well-known robustness of the perceptual grouping effect (e.g., Iversen 2008; Yoshida et al., 2010; Molnar et al. 2014; Molnar et al. 2016). As highlighted by Reviewer #2, having Spanish and Basque dominant “speakers as a sample equates that in Molnar et al. (2016), and thus overcomes the lack of direct behavioral evidence for a difference in rhythmic grouping across linguistic groups. Molnar et al. (2016)'s evidence on the behavioral effect is compelling, and the evidence on neural signatures provided by the present study aligns with it”. (iii) Regarding the fact that the “crucial effects are all mostly in the .01>P<.05 range”: we want to stress that the approach we used to detect the interaction effect was conservative, using a cluster-based permutation approach with no a priori assumptions about the location of the effect. The robustness of our approach has also been highlighted by Reviewer 2: “Data analyses. Sound, state-of-the-art methodology in the event-related field analyses at the sensor level.” In sum, despite some crucial effects being in the .01>P<.05 range, we believe that the statistical soundness of our analysis, combined with the lack of effect in the control condition, provides compelling evidence for our H1.

**Reviewer #3 (Recommendations For The Authors):**
Figures - Recommend converting all diagrams and plots to vector images to ensure they remain clear when zoomed in the PDF format.

Sure, thanks.

Figure 1: To improve clarity, the representation of sound durations in panels C and D should be revisited. The use of quavers/eighth notes can be confusing for those familiar with musical notation, as they imply isochrony. If printed in black and white, colour distinctions may be lost, making it difficult to discern the different durations. A more universal representation, such as spectrograms, might be more effective.

Thanks for the suggestion. It’s true that the quavers/eighth notes might be confusing in that respect. However, we find this notation as a relatively standard approach to define paradigms in auditory neuroscience, see for instance the two papers below. In the new version of the manuscript, we specified in the captions under the figure that the notes refer to individual tones, in order to avoid ambiguities.

- Wacongne, C., Labyt, E., Van Wassenhove, V., Bekinschtein, T., Naccache, L., & Dehaene, S. (2011). Evidence for a hierarchy of predictions and prediction errors in human cortex. Proceedings of the National Academy of Sciences, 108(51), 20754-20759.

- Dehaene, S., Meyniel, F., Wacongne, C., Wang, L., & Pallier, C. (2015). The neural representation of sequences: from transition probabilities to algebraic patterns and linguistic trees. Neuron, 88(1), 2-19.

Figure 2 : In panel C of Figure 2, please include the exact p-value for the interaction observed. Refrain from using asterisks or "n.s." and opt for exact p-values throughout for the sake of clarity.

Thank you for your suggestion. We have included the exact p-value for the interaction in panel C of Figure 2. However, for the remaining figures, we have chosen to maintain the use of asterisks and "n.s.". We would like our pictures to convey the key findings concisely, while the numerical details can be found in the article text. The caption below the image also provides guidance on the interpretation of the p-values: (statistical significance: **p < 0.01, *p < 0.05, and ns p > 0.05).

Figure 3 Note typo "Omission reponse"

Fixed. Thanks for noticing the typo.

A note: we moved the figure reflecting the main effect of long tone omission and the lack of main effect of language background (Figure 4 in the previous manuscript) in the supplementary material (Supplementary Figure 2).

References

Bendixen, A., SanMiguel, I., & Schröger, E. (2012). Early electrophysiological indicators for predictive processing in audition: a review. International Journal of Psychophysiology, 83(2), 120-131.

Heilbron, M., & Chait, M. (2018). Great expectations: is there evidence for predictive coding in auditory cortex?. Neuroscience, 389, 54-73.

Iversen, J. R., Patel, A. D., & Ohgushi, K. (2008). Perception of rhythmic grouping depends on auditory experience. The Journal of the Acoustical Society of America, 124(4), 22632271.

Molnar, M., Lallier, M., & Carreiras, M. (2014). The amount of language exposure determines nonlinguistic tone grouping biases in infants from a bilingual environment. Language Learning, 64(s2), 45-64.

Molnar, M., Carreiras, M., & Gervain, J. (2016). Language dominance shapes non-linguistic rhythmic grouping in bilinguals. Cognition, 152, 150-159.

Ross, J. M., & Hamm, J. P. (2020). Cortical microcircuit mechanisms of mismatch negativity and its underlying subcomponents. Frontiers in Neural Circuits, 14, 13.

Simon, J., Balla, V., & Winkler, I. (2019). Temporal boundary of auditory event formation: An electrophysiological marker. International Journal of Psychophysiology, 140, 53-61.

Studenova, A. A., Forster, C., Engemann, D. A., Hensch, T., Sander, C., Mauche, N., ... & Nikulin, V. V. (2023). Event-related modulation of alpha rhythm explains the auditory P300 evoked response in EEG. bioRxiv, 2023-02.

Yoshida, K. A., Iversen, J. R., Patel, A. D., Mazuka, R., Nito, H., Gervain, J., & Werker, J. F. (2010). The development of perceptual grouping biases in infancy: A Japanese-English cross-linguistic study. Cognition, 115(2), 356-361.

Zhang, Y., Yan, F., Wang, L., Wang, Y., Wang, C., Wang, Q., & Huang, L. (2018). Cortical areas associated with mismatch negativity: A connectivity study using propofol anesthesia. Frontiers in Human Neuroscience, 12, 392.

Ladinig, O., Honing, H., Háden, G., & Winkler, I. (2009). Probing attentive and preattentive emergent meter in adult listeners without extensive music training. *Music Perception*, *26*(4), 377-386.

Brochard, R., Abecasis, D., Potter, D., Ragot, R., & Drake, C. (2003). The “ticktock” of our internal clock: Direct brain evidence of subjective accents in isochronous sequences. *Psychological Science*, *14*(4), 362-366.

Potter, D. D., Fenwick, M., Abecasis, D., & Brochard, R. (2009). Perceiving rhythm where none exists: Event-related potential (ERP) correlates of subjective accenting. *Cortex*, *45*(1), 103-109.

Bouwer, F. L., Werner, C. M., Knetemann, M., & Honing, H. (2016). Disentangling beat perception from sequential learning and examining the influence of attention and musical abilities on ERP responses to rhythm. *Neuropsychologia*, *85*, 80-90.